# Disease Occurrence in- and the Transferal of Zoonotic Agents by North American Feedlot Cattle

**DOI:** 10.3390/foods12040904

**Published:** 2023-02-20

**Authors:** Osman Y. Koyun, Igori Balta, Nicolae Corcionivoschi, Todd R. Callaway

**Affiliations:** 1Department of Animal and Dairy Science, University of Georgia, Athens, GA 30602, USA; 2Bacteriology Branch, Veterinary Sciences Division, Agri-Food and Biosciences Institute, Belfast BT4 3SD, UK; 3Faculty of Bioengineering of Animal Resources, University of Life Sciences King Mihai I from Timisoara, 300645 Timisoara, Romania

**Keywords:** feedlot cattle, zoonoses, STEC O157:H7, *Salmonella*, *Escherichia coli*, *Campylobacter*, *Cryptosporidium*, *Brucella*, *Bacillus anthracis*, *Leptospira*

## Abstract

North America is a large producer of beef and contains approximately 12% of the world’s cattle inventory. Feedlots are an integral part of modern cattle production in North America, producing a high-quality, wholesome protein food for humans. Cattle, during their final stage, are fed readily digestible high-energy density rations in feedlots. Cattle in feedlots are susceptible to certain zoonotic diseases that impact cattle health, growth performance, and carcass characteristics, as well as human health. Diseases are often transferred amongst pen-mates, but they can also originate from the environment and be spread by vectors or fomites. Pathogen carriage in the gastrointestinal tract of cattle often leads to direct or indirect contamination of foods and the feedlot environment. This leads to the recirculation of these pathogens that have fecal–oral transmission within a feedlot cattle population for an extended time. *Salmonella*, Shiga toxin-producing *Escherichia coli*, and *Campylobacter* are commonly associated with animal-derived foods and can be transferred to humans through several routes such as contact with infected cattle and the consumption of contaminated meat. Brucellosis, anthrax, and leptospirosis, significant but neglected zoonotic diseases with debilitating impacts on human and animal health, are also discussed.

## 1. Introduction

Cattle, along with other ruminants, have provided humanity a stable supply of meat and dairy products since their domestication. In 2021, the per capita consumption of beef was approximately 26.7 kg in the United States [1], 16.9 kg in Canada [2], and 14.8 kg in Mexico [3]. North America is a large producer of beef for both domestic and export purposes, with more than 119 million heads of cattle, which represents approximately 12% of the world’s cattle inventory [4,5]. The United States has the largest cattle inventory (approximately 98.8 million cattle and calves in both beef and dairy operations) in North America [4,5]. Feedlots have been an integral part of modern beef cattle production in North America for more than 60 years, producing wholesome, highly desirable and marketable carcasses throughout the course of the year at a low cost to produce a high-quality protein food for humans [6,7]. Feedlots are typically located in the Great Plains region of North America and are located near both grain production and stocker/backgrounding regions. Cattle are fed in feedlots to take advantage of the economies of scale related to having many cattle located in one facility. 

Readily digestible, high-energy rations are provided to cattle through communal feed bunks or troughs (Figure 1) at feedlots (i.e., a confined area for growing or fattening cattle) during their final stage of growth, which is also known as finishing. It is at this point that marbling (i.e., intramuscular fat) is deposited in muscular tissues [6,7]. Feedlot rations mostly rely on corn (*Zea Mays L*.) supplemented with a protein source and often include by-products from other industries (e.g., dried distiller’s grains, brewer’s yeast) [6,8]. Cattle are usually fed 2–3 times per day in order to maximize feed consumption and growth efficiency. Feedlot cattle typically gain 1–2 kg/d and have a feed efficiency of approximately 5 to 6 kg feed/kg gain [9]. Commonly, these feedlot rations contain less than 10% forage (e.g., corn silage), and the feeding of such high-energy density rations can lead to the development of ruminal acidosis (low ruminal pH) [10]. When we feed cattle, we are actually feeding the microbial population of the rumen and hindgut (i.e., cecum, colon, and rectum), which ferment feedstuffs to produce Volatile Fatty Acids (VFAs) that cattle utilize for energy, and Microbial Crude Proteins (MCP), which ruminants use as their primary dietary protein source [11]. Feeding with starch has an advantage, as the microbial fermentation produces a greater proportion of propionate than when cattle are fed forage-based rations. Propionate is glucogenic and leads to intramuscular fat deposition (i.e., marbling) [12].

Despite ground-breaking advancements in the animal production and animal health aspects of feedlot systems, cattle can still have certain conditions and diseases that impact their health, growth performance, and carcass characteristics, and some of these can also impact human health [13,14,15,16]. Diseases are mostly transferred between cattle in a fecal–oral or direct contact fashion; however, they can originate from the environment and be spread by vectors (e.g., animals, rodents, or insects) or fomites (e.g., water, feed, surfaces, and soil), and pathogen carriage in the gastrointestinal tract (GIT) of cattle often leads to the direct or indirect contamination of feeds and the feedlot environment (e.g., water troughs and feed bunks, and feedlot pen surfaces) [14,15,17,18]. The circulation (and re-circulation) of pathogenic bacteria between different hosts, vectors, and the feedlot environment is ripe for the development of an on-farm endemic pathogen population that can impact both animal and human health. 

Amongst zoonotic pathogenic bacteria, foodborne pathogens such as *Salmonella* spp., Shiga toxin-producing *Escherichia coli* (STEC), and *Campylobacter* spp. are commonly associated with animal-derived foods and can be transferred to humans through several routes: (i) contact with positive cattle or carcasses, (ii) the consumption of contaminated or infected meat, and/or (iii) the consumption or irrigation of crops with water contaminated with cattle manure [13,19]. In addition, other zoonotic pathogenic agents with public health relevance such as *Cryptosporidium*, *Brucella*, *Bacillus anthracis*, and *Leptospira* and the diseases that they cause in humans are also discussed in this review. 

## 2. Structure of the North American Beef Industry

Beef cattle production in the United States is inextricably linked with the founding mythos of the Great Plains, or the “Old West”. Cattle ranchers from the frontier are often portrayed in movies and stories as independent and self-reliant heroes. Today’s North American cattle producers are heirs to this image and remain very independent and self-reliant. While increasing corporatization has impacted some segments of the cattle industry at the cow-calf production level, the beef industry of North America currently remains largely comprised of small producers. The beef industry has traditionally been highly decentralized and fragmented into five basic segments: cow-calf producers, stocker/backgrounder, feedlots, packers, and retail. The packer and retail segments are largely beyond the scope of this review, yet it is important to remember their role in the industry, which drives the production decisions made by ranchers for years before cattle reach the market. The beef production continuum is shown in Figure 2 and is best visualized as a pyramid in terms of the number of producers involved at each phase. However, an increasing degree of consolidation and vertical integration at the packing and retail levels has entered the beef production industry because there are fewer participants who can implement the required/suggested practices on the farms. This means that many of the practices that can implemented at larger, more well-funded production locations may not be implemented due to the economic and logistical constraints faced by the small producers. In the present review, we primarily focus on the live animal phases of beef production (Figure 2). 

### 2.1. Cow-Calf Producers 

Cow-calf producers are the foundation of cattle in the U.S. and are the most de-centralized phase of cattle production with thousands of producers scattered across the country, raising approximately 30 million calves each year. Cow-calf producers are often not able to be full-time cattle producers but must often work a “traditional job” (i.e., off-farm/ranch employment to generate a stable, consistent income) and must perform all of the farm tasks in their in their spare time, and as a result, many of their production decisions are driven by necessity, time availability, and logistics. This often limits the type of animal care procedures, as well as the procedures aimed at improving production efficiency, that can be implemented on any single farm. A typical beef producer in the southeastern United States is almost 60 years of age and works cattle on weekends and evenings when the weather and day-length allow. While most producers desire to maximize their profitability, many do not use the most up-to-date production methodologies (e.g., artificial insemination and estrus synchronization) due to the expense, time, and lack of skills and/or facilities involved. In general, producers attempt to calve in the spring and some use artificial insemination to improve their herd genetics and have a calf crop within a specified time window, with the majority utilizing a herd bull for ease of breeding.

Most cow-calf herds contain fewer than 50 cows, and these producers maintain a fairly stable herd size over the course of the year, marketing their calves themselves (from 180–240d of age, see Figure 3), often through local auction markets or sale barns [20,21]. When calves leave their farm of origin, they bring an “internal record of exposure and vaccination” with them in the form of their immune systems, which means that that while the calves are less susceptible to pathogens that they have previously been exposed to, they remain susceptible to novel pathogens (bacterial, protozoal, and viral). Stress acts as an immunosuppressant and is cumulative in its impacts. Calves at auction markets can undergo multiple simultaneous stresses from weaning and transport, as well as social stresses, and can therefore be moderately to severely immunosuppressed when commingled with calves from other farms. Collecting calves from multiple farms in a close-quarters environment is a recipe for disease amplification in a population of susceptible calves, including the spread of zoonotic pathogens within these calves, commingled with calves that originated from across broad geographic origins. 

Calves (weighing approximately 120–360 kg) typically remain at an auction market for 24 to 48 h before they are shipped to either a backgrounder/stocker facility or directly to a feedlot. The decision as to which pathway is utilized depends on calf size/age, breed, owner marketing strategies, and packer demands for quality or type of beef to be produced. Larger and older calves may be sent straight to a feedlot instead of to a background/stocker facility in order to begin the finishing process, but smaller calves may instead be sent to backgrounding/stocking to allow for slower growth and development. 

### 2.2. Backgrounders/Stockers and Feedlots

A tractor-trailer load of stressed and newly commingled calves is often transported for an additional 12 to 24 h (frequently transiting more than 1500 km in this time frame, whilst undergoing feed and water withdrawal, and often profound temperature changes) to a stocker or feedlot facility, which further exacerbates the susceptibility of these calves to disease exposure from cohorts. Upon arrival at either the feedlot or stocker facility, calves are typically rapidly vaccinated, identified, and allowed to rest and recuperate from the stresses of transport. These first days upon arrival are critical in setting cattle up for success as stresses can accumulate and result in the development of shipping fever in calves, which can impact morbidity and mortality among animals. Thus, it is critical to ensure that calves receive a ration designed to tempt them into beginning feed consumption quickly, in order to begin the supply of glucose, protein, and minerals to the immune system. Calves that are classified as “high risk” often require special care and added nutritional metaphylaxis and prophylaxis in the first few days after arrival in a stocker facility or feedlot. Stocker operators commonly feed native forages or crop residues (e.g., corn or wheat stubble) to cattle for 2–6 months in order to increase their growth and develop their frame (Figure 3). During backgrounding/stocking, cattle may consume protein or energy concentrates in their ration to increase their energy or protein intake; however, the amount of grain consumed in the stocker phase is typically much lower than that used in feedlots. The rations of stocker producers often contain by-products such as distiller’s grains, but mostly contain corn, with varying levels of processing (e.g., cracking or flaking) to improve its digestibility. When calves reach feedlot market weight (typically 270–370 kg), they are shipped to the feedlot for finishing or fattening prior to slaughter. 

In the feedlot, cattle are segregated in pens based on body weight, breed, sex, and special program enrollment (e.g., No Antibiotics Ever) and eat from communal feed bunks at the front of each pen. Cattle often enter the feedlot at approximately 350 kg and are fed diets containing a high Net Energy for Gain (NEG) concentration, which is achieved by feeding them diets rich in starch until they reach approximately 625 kg, the current market weight. The feeding/finishing period can last 90–300 d, depending on the size of the cattle when they enter the feedlot. 

Typically, according to the United States Department of Agriculture (USDA) animal census, there are more than 12 million cattle in U.S. feedlots at any time. While the vast majority of feedlot operations have a capacity of under 1000 heads, they only market a small percentage of the fed cattle to consumers. Feedlots with a capacity of more than 32,000 heads provide more than 40% of the fed cattle marketed [22]. Feedlots in the U.S. can reach a capacity of over 100,000 heads, which—assuming a 450 kg average weight for feedlot steers that consume 2% of their body weight (as dry matter (DM))—would require 9 kg (DM)/hd/d of feed, and a 50,000-head feedlot would require approximately 450,000 kg DM or 642,000 kg (as fed) of feed per day (approximately 7–8 train cars, or 20–25 tractor-trailer loads of feed). This typically requires feedlots to be largely self-contained facilities with an on-site feed mill (Figure 4). This means that many trucks bringing feed to each feedlot may take feed to other lots, and this represents a potential vector for zoonotic pathogens to be transmitted between feedlots. In addition, manure is often composted on site to mitigate the environmental impact and potentially generate a revenue stream by the sale of soil amendment for gardens; however, this can also carry zoonotic pathogens that can be transmitted to humans and other animals. It is clear that the infrastructure and activities needed to operate feedlots offer numerous opportunities for zoonotic pathogens to colonize and proliferate in cattle.

## 3. Zoonotic Agents with Public Health Relevance

There is a variety of pathogenic bacteria that are commonly found in cattle across North America. Most of these pathogens can (i) impact animal health; (ii) pose a threat to human health, such as foodborne pathogenic bacteria; and (iii) live in the GIT and are often undetected, as they may not cause illness in the host animal. This means that these pathogens may only be detected during the specific surveillance of cattle populations housed in a specific feedlot. Furthermore, many of these pathogenic bacteria can exist simultaneously in cattle, but little information currently exists on this issue of multiple pathogen colonization. Herein, we endeavor to discuss the most well-known human/animal threatening zoonotic agents of cattle with public health relevance. 

### 3.1. Salmonella spp.

*Salmonella enterica* serovars are one of the most important foodborne pathogens in North America, comprising more than 2500 serotypes that are often harbored in the GIT of a variety of animals such as mammals, birds, reptiles, and amphibians, as well as in a variety of different environments [14,23,24,25,26,27]. The major *Salmonella enterica* serovars associated with clinical infections in both cattle and humans are Dublin, Enteritidis, Heidelberg, Kentucky, Montevideo, Newport, and Typhimurium, and it should be noted that several of these serotypes can colonize the same animal simultaneously [14,25,28,29]. *S*. *Montevideo* was the most frequently reported serotype in North American cattle, while it was not one of the most frequently reported serotypes in other continents [23,28]. Moreover, *Salmonella* prevalence varies considerably by geographical region; a lower prevalence was recorded in the northern U.S. states and Canada than in southern states [30]. 

In the United States, non-typhoidal *Salmonella* is one of the most common bacterial foodborne diseases, resulting in an estimated 1.2 million domestically acquired foodborne infections, along with 450 deaths from approximately 130 outbreaks every year [19,29,31]. The infective dose for non-typhoidal *Salmonella* is reported at 10^3^ bacterial cells [30]. Salmonellosis in humans is often localized and self-limiting; however, severe cases require antimicrobial therapy and hospitalization [19,24,29,31]. Salmonellosis in humans is less associated with the consumption of beef or dairy products than compared to pork and poultry products [19,23,30]. However, certain cases have been traced back to cattle [19,23,30]. The contamination of lymph nodes that are processed into ground beef is one of the main ways for *Salmonella* spp. to enter the food chain [32,33].

The key transmission route of *Salmonella* in cattle is fecal–oral, and the prevalence of the pathogen in cattle varies, with reported estimates of 2–42% between-herd prevalence and 0–73% within-herd prevalence [14,34,35]. Cattle are asymptomatic carriers of *Salmonella* (i.e., a commensal of their GIT microbiota) [17,28] and can shed it at 10^3^ to 10^5^ CFU/g of feces, contaminating the farm environment and equipment [30,36]. It is believed that exposure to transport and lairage stress can increase the fecal excretion of *Salmonella* in feedlot cattle prior to slaughter [28,37]. The fecal shedding of *Salmonella* is subject to seasonal variation, reaching higher rates in the summer and early fall, declining in the winter months, and it has been reported that there is a correlation between shedding by animals and outbreaks in humans [14,17,32,38,39]. Although a physical correlation to ambient temperature has been observed, the internal temperature of the GIT is mostly stable; thereby, it seems that temperature is not the only source of the seasonality of *Salmonella* shedding through feces. Moreover, antimicrobial-resistant *Salmonella* (represented by varied serotypes such as *Salmonella* Newport, *Salmonella* Typhimurium, and *Salmonella* Reading) were detected in over 5000 individual fecal samples collected from multiple feedlots in the United States [40]. In Canada, the *Salmonella* prevalence in manure from feedlot cattle, beef carcasses, ground beef, and environmental samples is often reported to be low [13].

### 3.2. Shiga Toxin-Producing Escherichia coli (STEC) 

Shiga toxin-producing *E. coli* (STEC), also known as enterohemorrhagic *E. coli* (EHEC) or Vero toxin-producing *E. coli* (VTEC), are a family of zoonotic foodborne pathogens that can be naturally present in the GIT of cattle [41,42]. STEC that infect the human GIT are able to cause clinical symptoms ranging in severity from mild diarrhea to hemorrhagic colitis and life-threatening hemolytic uremic syndrome (HUS), a critical cause of acute renal failure in children [41,42]. STEC is characterized by a very low infective dose (<100 bacterial cells) in humans; however, hosts can asymptomatically harbor these pathogens as part of their GIT microbiota [43]. The frequency of STEC O157:H7 infections has been on the decline in North America over the past two decades due to improvements in meat safety, especially the implementation of “Test and hold” procedures for ground beef prior to shipment to consumers [44,45]. While most STEC-related illnesses have been often associated with the consumption of undercooked ground beef or through contaminated produce, pathogen transmission to humans can occur through contaminated drinking or recreational water, contact with cattle, pen surface contamination, and human-to-human contact [46,47].

Among STEC strains, enterohemorrhagic *E. coli* serotype O157:H7 has become one of the most important and well-studied pathogens as it frequently colonizes the GIT of cattle in North America [48,49,50]. While this is the most well-known and common STEC in North America, it is becoming clear that other STEC serotypes are impactful and play a role in human health. In the United States, along with O157, the top six non-O157 STEC serogroups (e.g., O26, O45, O103, O111, O121, and O145) have been recognized as adulterants in raw and non-intact ground beef [42,48]. This provides an economic incentive in addition to the ethical and moral incentives to reduce STEC contamination. The colonization and re-colonization of cattle with STEC occurs through fecal–oral contamination or the consumption of contaminated drinking water sources, or contaminated feeds, and the lower GIT of cattle, particularly the mucosa of the recto–anal junction (RAJ), is considered the major region for persistent colonization by *E. coli* O157:H7 [48,51,52]. STEC infections in cattle are usually asymptomatic, as they lack vascular receptors for the Shiga toxins (*Stx*), allowing this potent pathogen to thrive in the GIT while not causing damage to the host intestinal tissue or stimulating immune host defenses [42,47,53]. 

The levels of STEC O157:H7 in the GIT, digesta, and on hides of cattle prior to entering the commercial abattoir play a crucial role in the occurrence of carcass contamination during slaughter and processing [41,48,54]. Higher levels of STEC in cattle were correlated with higher carcass contamination levels. The previous literature showed that grain feeding increased the number of acid-resistant *E. coli* in feces of cattle, which has critical implications for food safety as the acid-resistance of the pathogen seems to be a factor in the transmission of this pathogen from cattle to humans [55]. In addition, STEC O157:H7 prevalence was increased in hide samples of cattle during transport (i.e., a common stressor to animals) from the feedlots to the abattoir and/or during lairage prior to slaughter [37,54,56]. Cattle that shed STEC O157:H7 at a rate of greater than 10^3^ or 10^4^ CFU/g of feces have been defined by the term “super-shedder”, and these high-shedding cattle remains the main vector of animal-to-animal transmission and production environment contamination [44,47,48,57]. STEC (*E. coli* O157 and non-O157) have been found in feedlot cattle feces and in feedlot environmental sources such as water troughs, lagoons, and soils in Canada [13,58]. Fecal prevalence rates of 0–79% have been reported for *E. coli* O157:H7 and 7–94% for the other ‘top six STEC’ (O26, O45, O103, O111, O121, and O145), and the prevalence is often higher during spring/summer than fall/winter [13,47,48,59,60]. It was reported that feedlot cattle farms can disseminate *E. coli* O157:H7 in the environment and that other animal vectors (e.g., feral swine), as well as flies, can contaminate leafy green vegetables on farms located in close proximity [46,61,62,63]. In North America, European starlings (*Sturnus vulgaris*) are considered a high-risk invasive bird species associated with the environmental dissemination of antimicrobial-resistant *E. coli* as these birds utilize feedlots during winter months for food sources [64]. Other studies have demonstrated that there is a potential spread of zoonotic pathogens to nearby fields and humans through dust spread from feedlot surfaces [61].

### 3.3. Campylobacter spp.

*Campylobacter* is one of the leading causative agents of bacterial foodborne gastro-enteritis in humans in the United States and Canada and can be transmitted to humans through human–animal contact (often via petting zoos), occupational exposure, the consumption of contaminated dairy (e.g., unpasteurized milk) and meat products, and contact with environment) [19,65,66,67,68]. *Campylobacter* is estimated to cause 1.3 million human illnesses every year in the United States [68], and the infection is often accompanied by abdominal pain and in some cases may lead to the development of the more severe Guillain–Barré syndrome in patients [69]. *Campylobacter* can also cause serious diarrhea in humans and has a very low infectious dose of as few as 500 organisms [67,68]. *Campylobacter jejuni* is the leading agent of reported human infections [65,67]. While poultry products are considered to be the leading source of human infections with *Campylobacter* in North America, cattle can serve as a vehicle for the transmission of this pathogen to humans [19]. Foodborne *Campylobacter* outbreaks in the United States (during 1998–2016) were attributed to dairy products (32%), chicken products (17%), and vegetables (6%), and more human outbreaks were reported during the summer (35%) than the spring (26%) and fall (22%) [67]. 

The colonization of *Campylobacter*, as a common commensal, in the GIT of cattle is significant not only regarding the potential for the contamination of the carcass at slaughter, but also regarding the environmental burden on farm and in transport through fecal shedding. It was reported that *Campylobacter* shedding by cows was 1.1 × 10^2^ CFU/g of feces, while shedding in calves was approximately 250-fold (2.7 × 10^4^ CFU/g of feces) more [30]. Studies conducted across the United States reported a *Campylobacter* prevalence ranging from 20 to 60% in feedlot and dairy cattle feces [70]. In particular, *C. jejuni* was detected in fecal samples collected from feedlots in the United States and Canada at a prevalence of 72–82% [13,65,66,70]. Up-to-date studies from Alberta, Canada, reported an increased antibiotic-resistant profile of fluoroquinolone-resistant *C. jejuni* isolates from around 1300 diarrheic patients connected to domestically acquired infections from cattle reservoirs [65]. Moreover, other researchers showed that, from 320 *C. jejuni* and 115 *C. coli* isolates collected from feedlot cattle farms in multiple states of the U.S., 35.4% of *C. jejuni* and 74.4% of *C. coli* isolates displayed increased fluoroquinolone resistance, which was remarkably higher than previously documented in United States reports [71]. *Campylobacter* species from feedlot manure runoff contaminates water supplies through agricultural runoff (due to rain events), posing serious human health concerns and increasing the risk of a waterborne outbreak [70]. Another important route of transmission is through migratory birds (e.g., European Starlings), and *Campylobacter jejuni* has been widely detected and identified in their feces [70].

### 3.4. Cryptosporidium spp.

Cryptosporidiosis is a disease in humans and cattle caused by a ubiquitous opportunistic enteric protozoan of the genus *Cryptosporidium*, is a global disease and one of the most common causes of diarrhea in both humans and livestock, and can be spread to humans from food animals and vice versa [72,73,74]. In cryptosporidiosis, parasite invasion and epithelial destruction in the small intestine by this causative agent results in crypt hyperplasia and apoptosis, villus atrophy and fusion, and physiological changes that impair intestinal nutrient absorption and cause diarrhea in the host [75,76]. Children, neonatal animals, and immunocompromised individuals are most susceptible to this parasitic disease, which is transmitted primarily through the fecal–oral route [74]. Contact with cattle, particularly with infected pre-ruminant calves, has been implicated as the root cause of many outbreaks in humans (e.g., veterinarians, researchers, and children attending agriculture-based activities and petting zoos) [74]. Moreover, food or water (e.g., lakes, rivers, and municipal drinking water without treatment) that is contaminated by cattle manure has been identified as a source of cryptosporidiosis outbreaks in humans [74,77,78]. The predominant *Cryptosporidium* species infecting humans are *C. parvum* and *C. hominis*, while *C. bovis*, *C. ryanae*, and *C. anderseni*, in addition to *C. parvum*, are the causative agents of bovine cryptosporidiosis [73]. 

In the United States and Canada, pre-weaned calves are considered important sources of zoonotic cryptosporidiosis transmission to humans. The previous literature documented that the prevalence of *Cryptosporidium* spp. between pre-weaned and post-weaned calves is age-related [79,80,81]. *Clostridium parvum*, the only prevalent zoonotic species in cattle, caused 85% of the *Cryptosporidium* infections in pre-weaned calves, while only 1% of the *Cryptosporidium* infections in post-weaned calves was due to this species [81]. In addition, a lower prevalence of cryptosporidiosis in 1–2-year-old dairy cattle (post-weaned calves and heifers) was found compared to younger (pre-weaned) calves [79,80]. Neonatal calves, which are not functional ruminants during the first 3–4 weeks of life, that are infected by *C. parvum* can suffer from serious scours (i.e., diarrhea with yellow pasty to watery feces) which can last up to 2 weeks and cause serious dehydration [72,82]. Infected calves can shed large numbers of infective oocysts in their feces, leading to environmental contamination and posing a threat to susceptible calves as well as humans [72,83]. Economic losses due to *Cryptosporidium* infections in neonatal calves are mostly associated with the cost of managing diarrheic animals, as well as mortalities [72,75]. Dehydration, weight loss, retarded growth performance, decreased feed efficiency, and losses due to mortality and morbidity are other repercussions of cryptosporidiosis, all of which leads to considerable economic losses [72,75].

### 3.5. Brucella spp.

Brucellosis, caused by *Brucella* spp., is a significant but neglected widespread bacterial zoonotic disease present around the world with debilitating impacts on human and animal health [84,85,86,87]. Humans are commonly infected through consuming adulterated unpasteurized/raw milk or dairy products [88,89,90,91]. However, direct contact with infected animals or their contaminated biological secretions (e.g., fetal or vaginal fluids and aborted fetuses or placentae), and exposure to anti-*Brucella* vaccines are other transmission routes of this occupational disease among animal handlers, veterinarians, and laboratory and abattoir personnel [90,92]. The inhalation of airborne agents was also reported as another transmission route of brucellosis in humans [87]. The use of personal protective equipment (PPE) to reduce the risk of brucellosis transmission is an effective measure among occupations that directly handle animals or their products [91]. Approximately 500,000 human brucellosis cases are reported each year to the World Health Organization (WHO), of which *Brucella melitensis* is the common causative agent [87,93]. The human brucellosis, also known as undulant fever or Malta fever, is characterized by non-specific clinical symptoms such as arthralgia, myalgia, sweats, miscarriage, abdominal pain, back pain, headache, profuse sweating, chills, and hepatomegaly [87,88,90]. Several countries in the world (located in the developed parts of Western and Northern Europe, Canada, Japan, Australia, and New Zealand) are free from the infectious agent [87,93]. Brucellosis is still endemic in Mexico, certain parts of Central and South America, the Mediterranean basin, the Middle East, India, and North Africa [89]. Nowadays, brucellosis in the United States is relatively rare (100–150 cases per year) and occurs more commonly in states that border Mexico (e.g., Texas and California) and in states where raw milk sale is legal [89,90,91,94]; a total of 75% of U.S. states allow different types of raw milk sales [89,90,91,94]. The incidence of human brucellosis in the United States has declined considerably over the years due to the successful U.S. State-Federal Cattle Brucellosis Eradication Program, as well as milk pasteurization [89,90]. 

Bovine brucellosis, caused by *B. abortus*, is a disease that occurs globally and causes substantial production loss along with a serious financial burden on producers [95]. The cattle farm environment is a convenient niche for brucellosis introduction, proliferation, and spread; improper biosecurity and management practices exacerbate the brucellosis progression in livestock animals [95]. The bacteria can live in soil, water, pasture, and manure for an extended time [96]. Therefore, the excretion of *Brucella* into the environment poses a risk to animal health [96]. In pregnant females, the primary symptom of brucellosis is abortion; however, the disease progression is often asymptomatic in young animals and non-pregnant females [97]. The bacterial agent can spread to multiple animals or herds through contaminated biological secretions such as fetal or vaginal fluids and aborted fetuses or placenta [92,97]. 

The smooth strain S19 and the rough strain RB51 vaccines are used in livestock for epidemiological control, yet both vaccines have disadvantages [90]. The RB51 strain, which has been used in the United States to vaccinate cattle against *B. abortus*, is virulent for humans (the infectious dose for *B. abortus* is 10–100 bacteria) and resistant to rifampin, a commonly used antibiotic used for treating human brucellosis [84,90,91]. Vaccinated animals can shed the strain into their milk; therefore, the presence and persistence of *Brucella* spp. in dairy products remain critical public health and food safety issues worldwide [90,91]. The contamination of the raw milk typically occurs either during milking or from the blood of infected animals being transferred to the milk [98]. Reportedly, animals infected with *B. abortus* can shed 10^3^ CFU/mL from blood to raw milk, yet supper-shedder hosts can shed even more (10^4^ CFU/mL) [97]. 

*Brucella* infections have been detected in varied terrestrial wild animals living in distinct environments (i.e., subtropical and temperate regions to arctic regions) [95]. The epidemiology of brucellosis in wildlife is often linked to the occurrence of the disease in livestock animals. Wild species can contribute to the re-introduction of *Brucella* agents along with infections in livestock (i.e., spillback) even in regions that are brucellosis-free or have had a successful eradication program [95]. Focusing on North America, bison, elk, and wild boars can become *Brucella* spp. reservoirs, and the latter two can spread the pathogenic agent to nearby cattle farms [95,98]. Brucellosis-impacted elk and bison populations from the Yellowstone Area in the United States have been shown to have a prevalence in the range of 35–60% [99].

### 3.6. Bacillus anthracis

Anthrax, known to humankind since ancient times, is a serious, naturally occurring, global zoonotic disease that affects domestic and wild animals, and directly/indirectly affects humans [100,101]. Anthrax is no longer considered a concern in developed countries due to effective control measures (e.g., vaccination, carcass disposal, and decontamination practices), yet it still occurs sporadically [101,102,103]. Anthrax is often found in agricultural regions of Central and South America, sub-Saharan Africa, central and southwestern Asia, southern and eastern Europe, and the Caribbean [101]. Over the years, there have been periodic outbreaks of anthrax in North America [102,103]. 

The causative agent of anthrax is *Bacillus anthracis*, an aerobic, Gram-positive, spore-forming, rod-shaped bacterium belonging to the *Bacillaceae* family [104,105]. In addition to causing naturally occurring anthrax, *B. anthracis* has been known to be a bioterrorism/agroterrorism weapon; therefore, surveillance systems have sought early detection of the disease [18,103]. The (dormant) spores produced by *B. anthracis* can persist in varied environments (e.g., soil, water, and animal hosts) for an extended time and are resistant to chemical and physical treatments such as radiation, desiccation, and heat application [104,105,106]. The spores enter the human body through varied routes and turn into active growing cells once the conditions are favorable, yet anthrax is not contagious [104,105,107,108]. The inhalation of spores from the hide or wool of infected animals, the ingestion of undercooked contaminated meat, skin abrasion, and, rarely, insect vectors (e.g., biting flies) are the main routes [104,105,107,108]. Reportedly, as few as 10 spores for herbivores and 200 to 55,000 spores for humans can be sufficient to cause an infection [109,110]. 

Anthrax in humans caused by the cutaneous transmission route accounts for approximately 95% of cases worldwide, due to the handling of carcasses and *B. anthracis*-contaminated by-products (e.g., hair, hides, and wool) of animals that were sick or died from the disease [18,105,107,108,111]. Animals often contract the disease through an oral ingestion of soil that is contaminated with spores [107,112]. It was reported that *B. anthracis* spores can survive in a soil environment for 300 years [107,112]. The most common clinical sign is a few sudden deaths in the herd without premonitory signs; bloating and hemorrhage from natural orifices (e.g., the nostrils, mouth, vulva, and anus) can be seen in dead animals [104,105,107,108]. 

In the United States, it was reported that *B. anthracis* spores can persist in alkaline soils present in the geographical corridor from Texas through Colorado, North and South Dakota to Montana, posing a primary risk for cattle and other herbivores [113,114,115]. In particular, a total of 63 anthrax cases in animals were confirmed in reference laboratories in Texas, the United States, during 2000–2018, and the last naturally occurring human case of cutaneous anthrax due to livestock exposure in Texas was reported in 2001 [111]. Texas experienced an increase in the number of animal anthrax cases in 2019 and state agencies suggested that more than 1000 animal losses might be attributed to the outbreak [111]. In Canada, repeated outbreaks in the wild bison populations still lead to concerns in the Northwest Territories, Northern Alberta, Manitoba, and Saskatchewan [116]. In 2006, an outbreak occurred in Saskatchewan and resulted in the loss of 804 livestock [117].

### 3.7. Leptospira spp.

Leptospirosis, caused by the spirochetal bacteria of the genus *Leptospira*, is considered one of the most widespread but neglected bacterial zoonotic diseases, affecting over 1 million humans globally every year with approximately 60,000 cases resulting in death [118,119,120]. Leptospirosis can cause a range of symptoms in humans, ranging from a mild fever, headache, and myalgia to more severe symptoms such as jaundice, renal failure, and multi-organ failure (i.e., known as Weil’s disease) that is primarily characterized by kidney and liver damage [118,119,120]. The disease is often misdiagnosed or even not recognized in humans as leptospirosis causes a myriad of symptoms that are also commonly displayed in many other diseases such as influenza and dengue fever, hampering the diagnosis accuracy of the disease in humans [118,119,120]. 

Leptospirosis is transmitted to humans by varied species of animals (e.g., cattle, sheep, pigs, horses, rodents, and dogs) through their infected urine as the bacteria can persist in the renal tubules of the host and are then excreted into (soil or water) environment through urination [121,122]. The bacteria can live in soil or water for an extended period of time, and humans can contract the disease through open wounds, conjunctiva, and mucous membranes when they are exposed to urine-contaminated soil or water [123,124]. Therefore, working in an abattoir or animal farms (i.e., occupational exposure) and swimming or wading in water bodies contaminated with urine (i.e., recreational exposure) are considered the main high-risk activities affecting the transmission course of leptospirosis in humans [118,119]. Approximately 100–150 human leptospirosis cases are reported every year in the United States, with Puerto Rico reporting the majority of the cases, followed by Hawaii [125]. In Mexico, during 2000–2010, there were over 1500 human leptospirosis cases reported (with 198 mortalities), and the majority of the cases were reported during the rainy season of the country [126]. 

Leptospirosis is a ubiquitous disease found in varied species of animals (e.g., cattle, sheep, pigs, horses, rodents, and dogs) and differs from human leptospirosis in terms of epidemiology, pathogenesis, clinical presentation, diagnosis, and control measures [122,127]. In particular, cattle are a common livestock reservoir and significantly impacted by varied *Leptospira* spp. that can cause abortion, neonatal illness, and reduced milk production in the hosts [122,127]. Bovine leptospirosis is commonly caused by three different serovars of *Leptospira*: *Leptospira borgpetersenii* serovar Hardjo (Hardjobovis), *Leptospira interrogans* serovar Hardjo (Hardjoprajitno), and *Leptospira interrogans* serovar Pomona [128,129,130]. Exposure to *Leptospira*-contaminated water sources, co-grazing with sheep, and the preference of natural service over artificial insemination are some of the major risk factors for leptospirosis disease in cattle [122,127]. Due to the colonization ability of *Leptospira* spp. in the renal tubes of cattle, bacterial shedding through urination into the environment can continue for an extended period of time and can also occur through semen and uterine discharges [128,131]. Vaccination strategies are used to prevent the shedding of leptospires in cattle urine [132,133]. According to a report by the USDA, based on a study conducted by National Animal Health Monitoring System (NAHMS), approximately one in five feedlots use vaccination to provide protection against leptospirosis in cattle [134].

## 4. Conclusions

Overall, there are many challenges that face producers of beef cattle in North America, including zoonotic pathogens that threaten both human and animal health. Zoonotic diseases are often transferred amongst pen-mates, but they can also originate from the environment and be spread by vectors (e.g., wild birds and insects) or fomites (e.g., animal contacting surfaces and airborne dust). Zoonotic pathogens such as *Salmonella*, Shiga toxin-producing *Escherichia coli*, and *Campylobacter* are commonly harbored in the GIT of cattle and are all too often associated with animal-derived foods as they can be transferred to humans through contact with infected cattle or carcasses, the consumption of contaminated or infected meat, and the consumption of water that is contaminated with cattle manure. The challenges posed by the presence of these pathogens as undetected passengers in the GIT of cattle are extensive and must be addressed in a holistic fashion. Furthermore, neglected but significant zoonotic agents such as *Cryptosporidium*, *Brucella*, *Bacillus anthracis*, and *Leptospira* still cause debilitating diseases in North American human populations that come in direct or indirect contact with cattle, cattle-surrounding environments, or cattle-originated biological materials, although relatively rarely compared to other parts of the world.

The beef cattle industry of North America has implemented numerous post-harvest pathogen reduction strategies, and has recently focused on on-farm or pre-harvest pathogen reduction strategies to improve human and animal health. It must be emphasized that these strategies must include non-antibiotic activities to avoid the development of antimicrobial/antibiotic resistance and improve the production efficiency or sustainability in order to ensure adoption by the industry. In addition, vaccination strategies have been used to provide protection against zoonotic diseases for several decades by the North American beef cattle industry.

## Figures and Tables

**Figure 1 foods-12-00904-f001:**
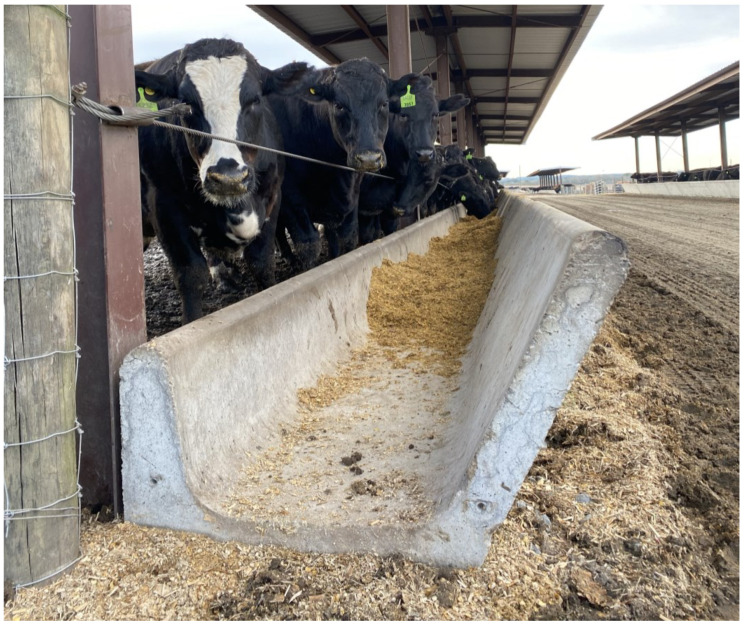
Cattle share communal feed bunks or troughs.

**Figure 2 foods-12-00904-f002:**
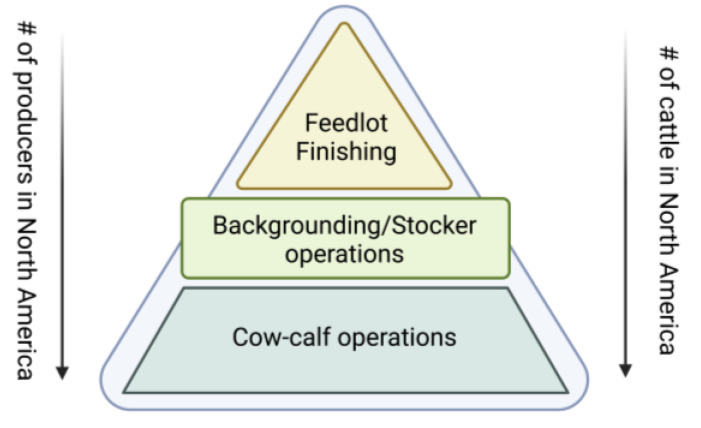
The beef production continuum visualized as a pyramid in terms of the number of producers involved at each phase.

**Figure 3 foods-12-00904-f003:**
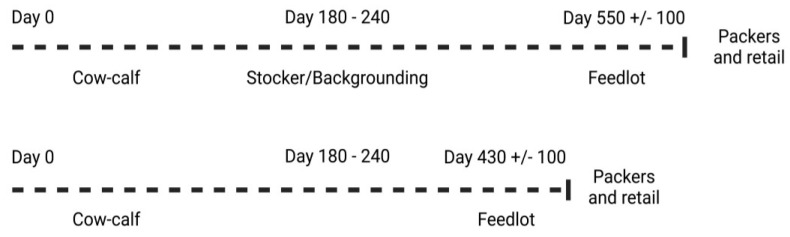
The beef industry has traditionally been fragmented into five segments: cow-calf producers, stockers/backgrounders, feedlots, packers, and retail. Created with BioRender.com.

**Figure 4 foods-12-00904-f004:**
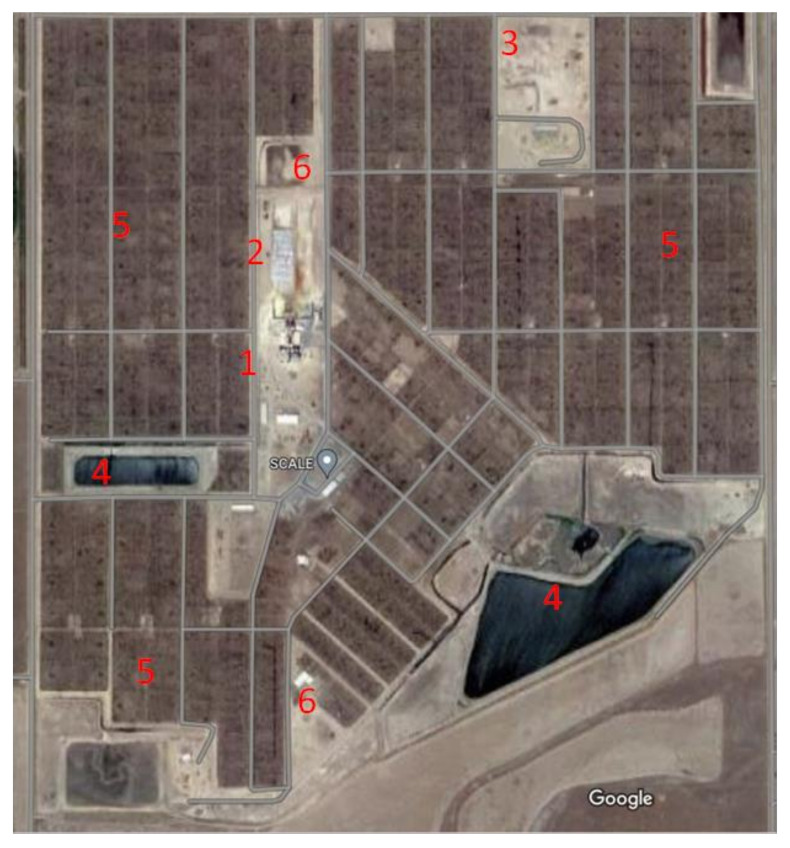
Aerial image of randomly chosen commercial feedyard. Feedmill is indicated by 1; silage pits are depicted by 2; manure/pen surface composting is tagged 3; 4 denotes water retention pond; 5 indicates cattle pens; and 6 highlights cattle working facilities. Image selected from Google Maps.

## Data Availability

Not applicable.

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
