# Peer review of "Disease Occurrence in- and the Transferal of Zoonotic Agents by North American Feedlot Cattle"

_foods, 2023, doi:10.3390/foods12040904_

Round 1

Reviewer 1 Report

This is a very nice review that provides a very useful summary of the North American cattle feeding industry and some associated zoonoses.

Just a few minor comments/corrections

Abstract - helpful if you could slip some mention of brucellosis into abstract. Have some suggestions for a bit that could be cut to help with this addition.

L14 can be rephrased as 'producing a high quality, wholesome...'

L19-21 This is too long of a sentence.  Would suggest dividing into two sentences.  End the first after pathogens on L 21.  The second sentence could be 'Pathogens with fecal-oral transmission cycle within feedyard populations...'

L45 Should be ' rations are'

L108 There are some cow-calf producers that derive their entire income from their cattle. They are not the norm, but this sentence should be  'often work a...'

L117 should be 'over the year'

L156 pens are often also divided by sex

L170 pen surfaces is a bit confusing as some pen surfaces are concrete. Best to say that composting the manure and avoid muddying things with pen surfaces.  Will need to change this terminology on Figure 4 as well.

L176 Data for figure 4 were taken from where?

L179  Not all pathogens of cattle are zoonoses. Best to say 'but some are human health threats...'

L205-209  Another excessively long sentence. Please divide into two.  Second sentence can start with 'However, '

L227 reported as low.

L237 has declined

L271  Need parallel construction here.  Would suggest 'transmission and production environment contamination.'

L288 What is professional exposure?  Need to clarify.

L305 and 306  You appear to be missing several superscripts here.

L386  would suggest 'is a disease...' 

Your section on Brucella is well done and useful as is too often ignored.

L405 can delete 'practices'

L414 is 'elk' as a plural.

L414 best to label as 'Conclusions'

Author Response

We would like to thank the reviewers and editors for their comments and suggestions.  They have strengthened the manuscript significantly.  We are very grateful for the time and effort you expended to make this better.  We have added the words necessary to reach the 6000 word mark requested by the editorial staff as well

Reviewer 1

Comments and Suggestions for Authors

This is a very nice review that provides a very useful summary of the North American cattle feeding industry and some associated zoonoses.

We thank the reviewer for their suggestions and kind words. 

Just a few minor comments/corrections

Abstract - helpful if you could slip some mention of brucellosis into abstract. Have some suggestions for a bit that could be cut to help with this addition.

Added as suggested

L14 can be rephrased as 'producing a high quality, wholesome...'

Changed as suggested

L19-21 This is too long of a sentence.  Would suggest dividing into two sentences.  End the first after pathogens on L 21.  The second sentence could be 'Pathogens with fecal-oral transmission cycle within feedyard populations...'

Changed as suggested

L45 Should be ' rations are'

Changed as suggested

L108 There are some cow-calf producers that derive their entire income from their cattle. They are not the norm, but this sentence should be  'often work a...'

Changed as suggested

L117 should be 'over the year'

Changed as suggested

L156 pens are often also divided by sex

Changed as suggested

L170 pen surfaces is a bit confusing as some pen surfaces are concrete. Best to say that composting the manure and avoid muddying things with pen surfaces.  Will need to change this terminology on Figure 4 as well.

Changed as suggested  Figure 4 was removed per other reviewer comments

L176 Data for figure 4 were taken from where? 

It was stated in the legend (Google Maps) but it has been removed per other comments of other reviewers

L179  Not all pathogens of cattle are zoonoses. Best to say 'but some are human health threats...'

Changed as suggested

L205-209  Another excessively long sentence. Please divide into two.  Second sentence can start with 'However, '

Changed as suggested

L227 reported as low.

Changed as suggested

L237 has declined

Changed as suggested

L271  Need parallel construction here.  Would suggest 'transmission and production environment contamination.'

Changed as suggested

L288 What is professional exposure?  Need to clarify.

Changed as suggested

L305 and 306  You appear to be missing several superscripts here.

Changed as suggested

L386  would suggest 'is a disease...' 

Changed as suggested

Your section on Brucella is well done and useful as is too often ignored.

Thank you!  We agree and appreciate that you see the lack of focus on it as important.  Especially in light of the US being declared Bangs free a month or so ago…..

L405 can delete 'practices'

Changed as suggested

L414 is 'elk' as a plural.

Changed as suggested

L414 best to label as 'Conclusions'

Changed as suggested

Reviewer 2 Report

Well written paper that only requires some minor editing as indicated on the marked pdf file.  

Author Response

We would like to thank the reviewers and editors for their comments and suggestions.  They have strengthened the manuscript significantly.  We are very grateful for the time and effort you expended to make this better.  We have added the words necessary to reach the 6000 word mark requested by the editorial staff as well

Reviewer 2

See attached PDF

It is difficult to respond pointwise to the PDF included comments.   But we thank the reviewer for marking these, we agreed with them all and in short, all responses are “Changed as suggested”

Reviewer 3 Report

Overall very good review.  My comments are minor. 

For consistency use feedlot or feedyard, but not both 

Same for gut or gastrointestinal tract 

Line 17 Anytime you use the word performance it is preferable to say "growth performance" as most people will not understand performance alone 

Line 21 lifestyle seems and odd choice of wording for pathogens, but I am not a microbiologist 

Line 52 the word approximately is not showing up correctly on my version, double check it 

Line 54 Change "does lead" to "can lead" to ruminal acidosis, this is highly managed in commercial feedlots by feeding 3X day and bunk management

Line 60 "forage-based" rations

Line 65 "cattle can still have certain conditions and diseases".... removing the word suffer, that has a negative connotation 

Line 66 growth performance 

Line 77 to 80 - Use positive vs. infected carcasses 

Line 83 and 84 - Cattle ranchers are self-reliant heroes 

Line 92 Replace farmers with ranchers or cattlemen, most cattle producers do not consider themselves farmers

Line 93 Consider replacing "live cattle production industry" with "beef industry" or "beef production continuum" 

Line 108 Add "typically" work a traditional job and maybe use the term off-farm income or off-ranch income 

Line 112 expense, time, lack of facilities, I don't think we can assume this is just expense 

Line 118 Prefer "auction market" vs. sale barn 

Line 124 Use the term commingled vs. mixed and change this sentence to "can be" instead of "are stressed" as not all calves from auction markets are stressed and high risk.... 

Can be moderately immunosuppressed 

Line 125 "in a new environment" a sale barn is stressful because it is new 

I like figure 3, wondering if you should "range" the final days?  Like up to 550 +/- 100 or something like that.  This does a good job representing total days in the system, but there is large variation and putting a +/- on the days would be helpful. 

Line 131 Not sure where the "less than 24 hours comes from" but based on my experience this can be 24 to 48 hours.  For example, if a sale is on Monday at noon, usually the cattle must be in place by 10 am Monday morning, but for people with "off-farm income" they drop the cattle off on Sunday afternoon.  Cattle are loaded on the truck Monday evening after the sale, last truck out at 7 pm Monday.  24 to 48 hours is a safer range here 

Line 131 - 132 The weight range here needs to be expanded, if you look at the TCFA auction market report, feeders steers are sold at up to 800 lb through auction markets 

Line 139 Use commingled not mixed 

Line 140 also water withdrawal, most cattle are dehydrated when arriving at feedlots 

Line 142 because of commingling

Line 151 Disagree that calves receive concentrate rations while in backgrounding or stocking, they will consume high-forage rations most likely, even if in a feedlot backgrounding operation the forage portion of the ration will be greater than the concentrate 

Line 152 Yes the feedlot rations will have byproducts, but most likely not "high" we want them to consume starch to produce marbling, so rewrite to say "often the rations in feedyards will contain some byproducts, but mostly contain corn, with varying levels of processing"

Line 156 in the feedyard we seldom know age, so they are usually sorted based on BW alone 

Line 159 Net energy for gain, not growth reword "fed diets containing a high net energy for gain concentration" 

Line 164 to 167 This is all true, but on a dry matter basis.  If on an as-fed basis this could be up to 642,000 kg of feed for a 70% DM diet, which is more train cars.  Just specify these cals are on a DM basis, because feed is delivered on an as-fed basis. 

Line 168-169 Most feedlots in the Southern great plains are not located near train lines and all commodities are delivered by truck.  For example, an average size feedlot down here may receive 20 to 25 loads of corn per day via truck (50,000 lb) 

Figure 4 - image 2 This looks like a pile not a pit of silage; image 4 should say water "retention" pond, I'm not convinced 6 is showing anything, but I cannot zoom in here.  The working facilities are likely the little silver structures located to the right of #5, below and to the right of #1, and above #1. Reword "dirt mounds" to soil mounds. 

Line 203 and sickness is dose dependent...?

LIne 216 and stress of lairage 

Line 261 prefer commercial abattoir vs. slaughterhouse 

Line 276 "top six STEC"

Line 278 Flies also according to Berry and Wells 

Line 305 and 306 1.1 x 10^2 and 2.7 x 10^4 

Author Response

We would like to thank the reviewers and editors for their comments and suggestions.  They have strengthened the manuscript significantly.  We are very grateful for the time and effort you expended to make this better.  We have added the words necessary to reach the 6000 word mark requested by the editorial staff as well

Reviewer 3

Overall very good review.  My comments are minor. 

For consistency use feedlot or feedyard, but not both 

You’re right.  It’s such a regional variation that I was trying to cover both bases, but consolidated now into feedlot  Changed as suggested

Same for gut or gastrointestinal tract 

Changed as suggested

Line 17 Anytime you use the word performance it is preferable to say "growth performance" as most people will not understand performance alone 

Changed as suggested

Line 21 lifestyle seems and odd choice of wording for pathogens, but I am not a microbiologist 

I agree it seems odd, but it’s basically how we are trying to view the microbial ecology in the same lens system as macroecological ones….that helps most people understand better what these organisms are trying to do

Line 52 the word approximately is not showing up correctly on my version, double check it 

Weird.  Changed as suggested.  Great Catch!

Line 54 Change "does lead" to "can lead" to ruminal acidosis, this is highly managed in commercial feedlots by feeding 3X day and bunk management

Changed as suggested

Line 60 "forage-based" rations

Changed as suggested

Line 65 "cattle can still have certain conditions and diseases".... removing the word suffer, that has a negative connotation 

Changed as suggested

Line 66 growth performance 

Changed as suggested

Line 77 to 80 - Use positive vs. infected carcasses 

Changed as suggested

Line 83 and 84 - Cattle ranchers are self-reliant heroes 

You’re absolutely right.  I was hesitant to be that critical, but since you agree…..Changed as suggested

Line 92 Replace farmers with ranchers or cattlemen, most cattle producers do not consider themselves farmers

Changed as suggested

Line 93 Consider replacing "live cattle production industry" with "beef industry" or "beef production continuum" 

Changed as suggested

Line 108 Add "typically" work a traditional job and maybe use the term off-farm income or off-ranch income 

Changed as suggested

Line 112 expense, time, lack of facilities, I don't think we can assume this is just expense 

Changed as suggested

Line 118 Prefer "auction market" vs. sale barn 

Changed as suggested

Line 124 Use the term commingled vs. mixed and change this sentence to "can be" instead of "are stressed" as not all calves from auction markets are stressed and high risk.... 

Can be moderately immunosuppressed 

Changed as suggested

Line 125 "in a new environment" a sale barn is stressful because it is new 

Changed as suggested

I like figure 3, wondering if you should "range" the final days?  Like up to 550 +/- 100 or something like that.  This does a good job representing total days in the system, but there is large variation and putting a +/- on the days would be helpful. 

Great points, Changed as suggested

Line 131 Not sure where the "less than 24 hours comes from" but based on my experience this can be 24 to 48 hours.  For example, if a sale is on Monday at noon, usually the cattle must be in place by 10 am Monday morning, but for people with "off-farm income" they drop the cattle off on Sunday afternoon.  Cattle are loaded on the truck Monday evening after the sale, last truck out at 7 pm Monday.  24 to 48 hours is a safer range here 

Great point, and I agree Changed as suggested

Line 131 - 132 The weight range here needs to be expanded, if you look at the TCFA auction market report, feeders steers are sold at up to 800 lb through auction markets 

You’re right, I kept coming back to southeastern calves too much.  Changed as suggested

Line 139 Use commingled not mixed 

Changed as suggested

Line 140 also water withdrawal, most cattle are dehydrated when arriving at feedlots 

Changed as suggested

Line 142 because of commingling

Changed as suggested

Line 151 Disagree that calves receive concentrate rations while in backgrounding or stocking, they will consume high-forage rations most likely, even if in a feedlot backgrounding operation the forage portion of the ration will be greater than the concentrate 

Changed as suggested

Line 152 Yes the feedlot rations will have byproducts, but most likely not "high" we want them to consume starch to produce marbling, so rewrite to say "often the rations in feedyards will contain some byproducts, but mostly contain corn, with varying levels of processing"

Changed as suggested

Line 156 in the feedyard we seldom know age, so they are usually sorted based on BW alone 

Changed as suggested

Line 159 Net energy for gain, not growth reword "fed diets containing a high net energy for gain concentration" 

Changed as suggested

Line 164 to 167 This is all true, but on a dry matter basis.  If on an as-fed basis this could be up to 642,000 kg of feed for a 70% DM diet, which is more train cars.  Just specify these cals are on a DM basis, because feed is delivered on an as-fed basis. 

Great catch! Thank you  Changed as suggested

Line 168-169 Most feedlots in the Southern great plains are not located near train lines and all commodities are delivered by truck.  For example, an average size feedlot down here may receive 20 to 25 loads of corn per day via truck (50,000 lb) 

Changed as suggested

Figure 4 - image 2 This looks like a pile not a pit of silage; image 4 should say water "retention" pond, I'm not convinced 6 is showing anything, but I cannot zoom in here.  The working facilities are likely the little silver structures located to the right of #5, below and to the right of #1, and above #1. Reword "dirt mounds" to soil mounds. 

Figure 4 has been removed due to concerns from reviewers, but thanks for the suggestions. You’re right

Line 203 and sickness is dose dependent...?

Changed as suggested

LIne 216 and stress of lairage 

Changed as suggested

Line 261 prefer commercial abattoir vs. slaughterhouse 

Changed as suggested

Line 276 "top six STEC"

Changed as suggested

Line 278 Flies also according to Berry and Wells 

Changed as suggested

Line 305 and 306 1.1 x 10^2 and 2.7 x 10^4

Changed as suggested